# Audio Large Language Models Can Be Descriptive Speech Quality Evaluators

**Chen Chen**[†,1,2]   **Yuchen Hu**[1]   **Siyin Wang**[3]   **Helin Wang**[4]   **Zhehuai Chen**[2]
**Chao Zhang**[†,3]   **Chao-Han Huck Yang**[†,2]   **Eng Siong Chng**[†,1]
[1]Nanyang Technological University   [2]NVIDIA
[3]Tsinghua University   [4]Johns Hopkins University
[†]corresponding authors: {cchen1,hucky}nvidia.com; ASESChng@ntu.edu.sg

## Abstract

An ideal multimodal agent should be aware of the quality of its input modalities. Recent advances have enabled large language models (LLMs) to incorporate auditory systems for handling various speech-related tasks. However, most audio LLMs remain unaware of the quality of the speech they process. This limitation arises because speech quality evaluation is typically excluded from multi-task training due to the lack of suitable datasets. To address this, we introduce the first natural language-based speech evaluation corpus, generated from authentic human ratings. In addition to the overall Mean Opinion Score (MOS), this corpus offers detailed analysis across multiple dimensions and identifies causes of quality degradation. It also enables descriptive comparisons between two speech samples (A/B tests) with human-like judgment. Leveraging this corpus, we propose an alignment approach with LLM distillation (ALLD) to guide the audio LLM in extracting relevant information from raw speech and generating meaningful responses. Experimental results demonstrate that ALLD outperforms the previous state-of-the-art regression model in MOS prediction, with a mean square error of 0.17 and an A/B test accuracy of 98.6%. Additionally, the generated responses achieve BLEU scores of 25.8 and 30.2 on two tasks, surpassing the capabilities of task-specific models. This work advances the comprehensive perception of speech signals by audio LLMs, contributing to the development of real-world auditory and sensory intelligent agents.

## 1 Introduction

In recent years, large language models (LLMs) have exhibited impressive abilities as general-purpose NLP task solvers (Brown et al., 2020; Wei et al., 2022; Touvron et al., 2023). Meanwhile, researchers are actively exploring the extension of LLM capabilities to handle the speech processing task by integrating acoustic information into pre-trained LLMs (Fathullah et al., 2024). This endeavour has resulted in the emergence of increasing audio large language models that can simultaneously process both audio and text inputs (Tang et al., 2021; Bapna et al., 2022; Rubenstein et al., 2023; Chu et al., 2023; 2024; Tang et al., 2023; Chen et al., 2024; Yang et al., 2024a). Through supervised cross-modalities alignment, these models can handle increasingly intricate tasks based on the bi-modal comprehension.

Existing efforts of audio-text alignment in LLMs mainly focus on content information–refers to the explicit meaning and linguistic structure (Radhakrishnan et al., 2023; Fathullah et al., 2024; Tang et al., 2024). However, human speech encompasses a wealth of information beyond words, such as emotion, accent, and timbre. More recent research (Ao et al., 2024; Lin et al., 2024b) introduces the importance of this paralinguistic information during LLMs integration, emphasizing their critical role in spoken dialogue (Lin et al., 2024a). Nevertheless, current audio LLMs may overlook the intrinsic quality of human speech as a form of signal, like distortion, noisiness, and coherence. These undiscovered capabilities lead to an intriguing phenomenon: while these models are robust enough to extract useful information from diverse speech inputs, however, they remain *unaware* of the quality of the speech signal itself.

To evaluate speech quality, the overall Mean Opinion Score (MOS) (Viswanathan & Viswanathan, 2005) is typically regarded as an important indicator in modern communication networks. Plenty of deep neural networks are dedicated to predicting the average MOS as a regression task (Lo et al., 2019; Choi et al., 2021). However, subjective ratings exhibit significant variability, and the annotations in existing datasets show a non-negligible variance (Cooper & Yamagishi, 2021; Wu et al., 2024b). More importantly, simply predicting a numerical MOS is overly simplistic and provides no insight into the underlying causes of quality estimation (Mittag, 2021). Motivated by this, we aim to teach LLMs to evaluate the quality of speech like humans, which are expected to provide *descriptive analysis* and *reasonable judgement*. We hereby highlight the significance of this understanding capability, which can be used to automate the evaluation of performance in modern generative systems, such as text-to-speech or speech editing models. Moreover, unifying understanding and generation tasks within a single Transformer based model has emerged as a notable trend in academic research (Zhang et al., 2023; Défossez et al., 2024; Wu et al., 2024d). From this perspective, it becomes increasingly crucial for a model that knows the quality of input, which potentially enables it to engage in a self-improvement loop as an agent.

To the best of our knowledge, existing human speech quality datasets consist solely of numerical scores, and do not include any natural language-based descriptions or analyses. In this work, we first bridge this gap by introducing a new dataset comprising natural language descriptions generated based on authentic human ratings of multidimensional speech quality assessment corpus (Mittag et al., 2021). Specifically, for each speech sample, we leverage the meta-information from the corpus, prompting LLMs to generate an aligned analysis of its multi-dimensional characteristics with the help of demonstrations, followed by reasoning and a final overall MOS rating. For example: "*This given speech has very slight distortion, without any background noise. However, there is a noticeable discontinuity that significantly influences its perceptive quality. Taking into account all factors, the overall MOS score is only 2.4.*" In this context, an A/B test dataset is also composed with a similar strategy. We sample two speech segments and task the LLM with performing a descriptive comparison of their relative strengths and weaknesses across specific sub-dimensions, culminating in a well-justified preference judgment, as illustrated in Fig. 2.

Given this dataset, the audio-interfacing LLMs are expected to generate the same response based solely on the raw audio input. The challenge of this task lies in requiring the audio LLMs to automatically focus on sub-dimensional information within the speech signal, and then provide cross-modal response with reasoning. Accordingly, we propose an effective learning strategy called ALLD, which aligns the generated sequence of the audio LLM to an expert LLM's response based on meta information. Specifically, the output of audio LLM and expert LLM are formulated as preferred-dispreferred completions for preference optimization algorithms (Rafailov et al., 2024). Notably, the expert LLM is exceptionally set as a reference model for token-level distillation, which is distinct from the stereotype of the reference model in mainstream reinforcement learning from human feedback strategy. Experimental results demonstrate the efficacy of ALLD that surpasses the previous state-of-the-art regression model, achieving a mean square error of $0.17$ on MOS prediction and $98.6\%$ accuracy on the A/B test. Furthermore, ALLD effectively enhances the language capability of the audio LLM in terms of BLEU, which is often degraded during its audio-text pre-training. The learned ability of quality evaluation can also be applied to unseen speech domains.

In summary, our contributions are in three folds: (i) We direct our research focus on the inability of audio LLMs to perceive the quality of input speech, highlighting the importance of speech quality evaluation for multimodal agents. (ii) The first descriptive speech quality evaluation dataset is introduced to bridge the speech evaluation gap for audio LLMs. Beyond the commonly used MOS scores, this dataset provides natural language-based multidimensional descriptions of sound characteristics and analysis of quality degradation. Furthermore, it extends to comparative tasks like A/B tests, which require analysis and judgment by audio LLMs. (iii) We propose a learning strategy called ALLD that enables audio LLMs to achieve end-to-end perception and generation. By employing token-level distillation, ALLD effectively mitigates the language capability degradation of audio LLM during its pre-training. Consequently, it improves the quality of generated responses in terms of BLEU and also surpasses traditional SOTA regression models in terms of multiple metrics.

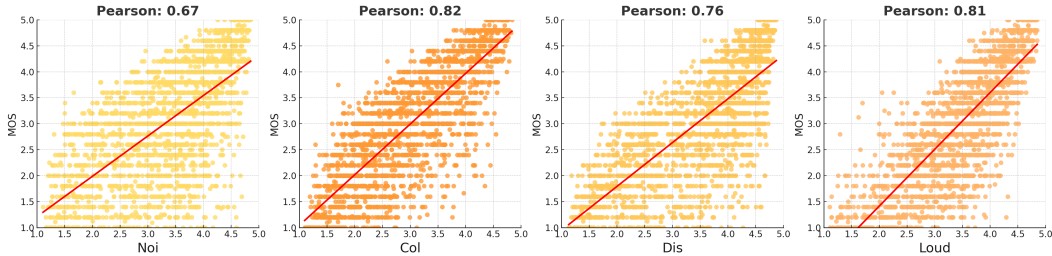

Figure 1: The relationship between MOS and four sub-dimensions: *Noisiness*, *Coloration*, *Discontinuity*, and *Loudness*. The red line represents the linear regression line fitted to the data points, showing the linear trend between each metric and MOS. The Pearson correlation coefficient is displayed above each plot, indicating the strength of the linear relationship.

## 2 RELATED WORK

### 2.1 LLMs FOR AUDIO INFORMATION PERCEPTION

In recent years, extending LLMs with an audio encoder has successfully demonstrated the ability to perceive audio (Wang et al., 2023b), as known as audio LLM, establishing a versatile framework capable of handling various audio-to-text tasks (Wang et al., 2024a; Yang et al., 2024b). Based on semantic alignment, the downstream tasks cover speech recognition, speech translation, question-answering (Cheng et al., 2023), and spoken language understanding (Li et al., 2024b), etc. These tasks can be solved by cascaded recognition modules and text-based LLMs. Beyond the linguistic content, existing efforts investigate the paralinguistic information in speech, like emotion (Santoso et al., 2024) and speaker attribution (Wu et al., 2024a), which plays an important role in spoken dialogue. Meanwhile, LLMs with audio encoder framework are also well-suited for audio scene and music understanding (Zhou et al., 2024), which can even leverage the reasoning capabilities of LLMs to perform the audio-based reasoning (Li et al., 2024a). Broadly speaking, speech-text foundation models adopt multitasking strategies with multi-task training, as seen in models like AudioPaLM (Rubenstein et al., 2023), SALMONN (Tang et al., 2023), Qwen-Audio (Chu et al., 2023; 2024), WavLLM (Hu et al., 2024), and SpeechVerse (Das et al., 2024).

### 2.2 SPEECH QUALITY EVALUATION

Mainstream efforts for speech quality evaluation primarily focus on MOS prediction–the subjective score usually used for speech quality evaluation (Streijl et al., 2016). Existing corpus collect diverse speech from real audio scenarios or TTS systems and score them from 1 to 5 by different independent human listeners (Cooper & Yamagishi, 2021; Mittag et al., 2021; Maniati et al., 2022), the various neural networks are proposed to estimate the MOS score in a regression manner (Lo et al., 2019; Choi et al., 2021; Cooper et al., 2022). More recently, self-supervised learning models like WavLM (Chen et al., 2022) have contributed to improving the accuracy since they can extract better speech representation from raw speech. The potential of audio LLMs to evaluate speech quality is also investigated in Wang et al. (2024c); Zezario et al. (2024). Additionally, considering the variation of different human listeners, only predicting the MOS score can seem somewhat limited. More recent work (Wu et al., 2024b;c) has proposed going beyond just predicting the MOS mean by also accounting for the uncertainty introduced by different raters' subjective preferences.

**Summary.** Despite the remarkable progress of MOS prediction by regression models, there is no existing dataset that contains natural language to describe or evaluate speech quality, which results in audio LLMs overlooking this task during their multi-task training. This paper aims to address this gap by introducing both a dataset and learning methods that enable audio LLMs to describe and evaluate speech quality in a human-like manner, thereby enhancing the multimodal system's comprehensive perception and understanding of speech signals.

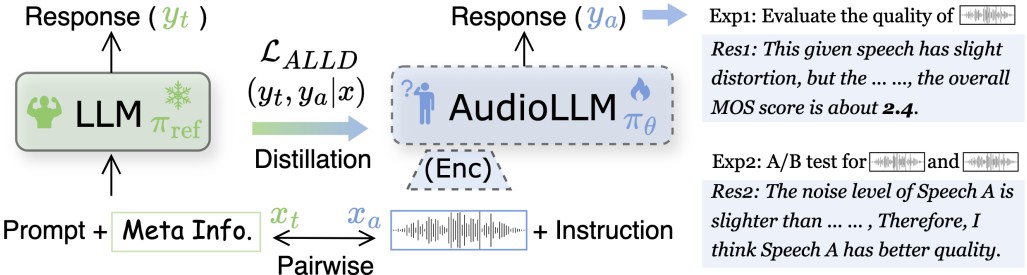

Figure 2: The framework of ALLD and training examples. "Meta info." is the multi-dimensional ratings annotated by human listeners for the pairwise speech sample. ALLD aims to align the audio LLM response $y_a$ to $y_t$ via token-level distillation, where $\pi_{\text{ref}}$ is exceptionally set as an expert LLM.

## 3 BACKGROUND

**Audio LLMs.** Based on whether the acoustic representation is continuous or discrete, audio LLMs can be classified into two categories: (i) One approach uses an audio codec to discretize the input, expanding the LLM's vocabulary to facilitate interactions and understanding between audio and text (Zhang et al., 2023). (ii) The other approach involves using a pre-trained encoder, such as an ASR model's encoder or a self-supervised learning model, to process raw waveforms. The output is subsequently combined with the LLM's word embeddings via a modality adapter, facilitating the integration of both audio and text information (Bapna et al., 2022; Chu et al., 2024). In this work, we address the speech evaluation task by utilizing the second type of audio LLM as our chosen model for this purpose. A trainable encoder could ensure that the relevant features are effectively extracted from the raw audio signal, as illustrated in Figure 2.

**Speech Quality Description.** To give the insight into speech quality multidimensional space, Wältermann defines three orthogonal dimensions for modern communication networks, namely *Noisiness*, *Coloration*, and *Discontinuity*. Later, *Loudness* was added as a fourth dimension in (Côté et al., 2007), although it is *not* entirely orthogonal to the other dimensions. Through audio listening tests, these four dimensions can be quantified from 1 to 5 like MOS, by the subjective perception of the human listener. To validate their impact on MOS score, we employ the human-annotated data from NISQA (Mittag et al., 2021), and visualize the scatter plot based on 2.5k data points. The Pearson correlation coefficient with linear regression is shown in Figure 1. These results reveal that all four metrics contribute meaningfully to the overall MOS, with *Coloration* (0.82) and *Loudness* (0.81) playing the most influential roles in determining speech quality.

## 4 METHODOLOGY

### 4.1 DATASET GNERATION

**MOS Prediction.** To generate a descriptive training corpus, we leverage the reasoning and language generation capabilities of LLMs. Given a tuple of meta information $x_t = \{\text{mos, noi, dis, col, loud}\}$, we design the task prompt including an illustration of each factor and the requirement of generative content. Considering the strong linear relationship between them and MOS, LLMs should highlight those prominent factors and analyze their impact on overall MOS. For example, in TTS-synthesized speech, it is common to encounter distortion in specific words while the other three metrics remain acceptable. In such cases, colouration ('col') becomes the key factor that needs to be emphasized, and the model should explicitly state its negative impact before estimating the final MOS rating.

In practice, we found that existing open-source LLMs, even at the 70B scale, struggle to follow such detailed instructions effectively. To address this, we optimize the data generation pipeline using *in-context learning*. Specifically, we select representative demonstrations with different MOS levels and manually write corresponding responses. Typically, 3 to 5 demonstrations are sufficient to significantly improve the quality of the model's outputs. The relevant prompts, demonstrations, and generated samples are provided in Appendix B for reference.

**A/B Test** is a common subjective evaluation method where human listeners are asked to compare two speech samples and select the better one with higher quality. Typically, A/B tests are conducted to compare different TTS systems or to contrast synthesized speech with real human speech. To mimic the human judgments, we hope the audio LLM to automatically capture sub-dimensional distinctions and then accordingly make the decision of preference. Similar to MOS prediction, we employ specific task prompts and demonstrations to guide the LLM's generation. The model first provides a detailed comparison of the two speech samples, followed by a well-reasoned preference judgment, as illustrated in Figure 2.

In addition to the above utterance-level task, we further introduce a word-level task for audio LLMs, namely synthetic word detection (**SWD**). Specifically, given an unlabeled speech input, the current spoof detection task aims to identify which speech frames are synthetic Li et al. (2024a), while the SWD task asks the audio LLM to reason which specific words are synthetic. This task is designed to address the increasingly sophisticated capabilities of speech editing models. In recent work such as VoiceCraft (Peng et al., 2024), human listeners have become almost incapable of distinguishing between original and edited speech samples. The detailed generation pipeline is in Appendix B.

## 4.2 ALIGNMENT WITH LLM DISTILLATION

**Can In-context learning handle speech quality evaluation?** We first explored whether gradient-free methods like in-context learning could prompt audio LLMs to perceive speech quality. To simplify the setting, the ICL template is set as: "*Please evaluate the noisiness of the speech by predicting 'clean' or 'noisy'. For example: [audio1] is noisy, and [audio2] is clean. [audio3] is:*". We try different prompt formats, demonstrations, sub-dimensions of speech quality, and audio LLMs (listed in Appendix B). The conclusion is that existing open-source audio LLMs are unable to perform speech evaluation effectively using this method, often exhibiting widespread hallucinations. An underlying reason is that the supervised finetuning of the audio tasks leads to significant degradation of LLMs' original capabilities on instruction-following and reasoning (Wang et al., 2023a), thus hindering their generality in handling unseen tasks.

**ALLD.** To enhance the generation quality, we introduce an effective distillation strategy that aligns the outputs of audio LLM and LLM, as shown in Fig 2. Given the audio input $x_a$, the response $y_a$ of audio LLM $\pi_\theta$ is expected to be aligned with reference response $y_t$. Considering the degradation of $\pi_\theta$, expert LLM $\pi_{\text{ref}}$ is involved as a reference model based on meta information $x_t$. Inspired by recent reinforcement learning from human feedback (RLHF) algorithms (Ouyang et al., 2022), we formulate the alignment objective of $\pi_\theta$ with a learned reward function $r_\phi(x, y)$ as:

$$\max_{\pi_\theta} \mathbb{E}_{(x_a, x_t) \sim \mathcal{D}, y \sim \pi_\theta(y|x_a)}[r_\phi(x_a, y)] - \beta D_{\text{KL}}(\pi_\theta(y|x_a) \| \pi_{\text{ref}}(y|x_t)) \tag{1}$$

Notably, here $\pi_{\text{ref}}$ can be viewed as a teacher LLMs, providing **token-level distillation** to the $\pi_\theta$ in the form of a KL-divergence constraint. In contrast, mainstream RLHF algorithms typically set the $\pi_{\text{ref}}$ as a copy of frozen $\pi_\theta$, to prevent the model from making drastic changes. For $r_\phi(x, y)$, we utilize the implicit reward modelling proposed in DPO (Rafailov et al., 2024), where $y_t$ and $y_a$ are formulated as preferred-dispreferred completions for Bradley-Terry model. After sampling enough $y_a$, a dataset of comparisons $\mathcal{D} = \{x_a^{(i)}, x_t^{(i)}, y_a^{(i)}, y_t^{(i)}\}_{i=1}^N$ is composed for preference optimization, and the training objective can be re-written as:

$$\mathcal{L}_{\text{ALLD}}(\pi_\theta; \pi_{\text{ref}}) = -\mathbb{E}_{(x, y_a, y_t) \sim \mathcal{D}} \left[ \log \sigma \left( \beta \log \frac{\pi_\theta(y_t|x)}{\pi_{\text{ref}}(y_t|x)} - \beta \log \frac{\pi_\theta(y_a|x)}{\pi_{\text{ref}}(y_a|x)} \right) \right] \tag{2}$$

It is noted that $x_a$ and $x_t$ are uniformly denoted as $x$, since they carry equivalent information ($x_t$ is embedded within the audio $x_a$). In Eq. 2, Audio LLM $\pi_\theta$ is optimized to extract information from $x_a$ and predict the consistent sequence $y_a$ with $y_t$. This sampling-optimization strategy can alleviate the exposure bias problem commonly seen in typical SFT, as discussed in previous works (Bahdanau et al., 2016; **?**).

In practice, we utilized a smaller LLM (e.g., Qwen-7B) as $\pi_{\text{ref}}$ to reduce the computational cost, while its tokenizer keeps consistent with $\pi_\theta$ for distillation. Furthermore, ALLD is theoretically iterable—$y_a$ can progressively approach $y_t$ with a repeatable sampling-learning loop. To simplify the training process, we sample $y_a$ only once for preference optimization. However, the sampling of $y_a$ requires a warm-up SFT on a subset of $\mathcal{D}$, due to the lack of zero-shot capability of the $\pi_\theta$, as mentioned at the beginning of this section.

## 5 EXPERIMENTAL RESULT

### 5.1 SETUP

**Dataset**. We used the NISQA (Mittag et al., 2021) that contains more than $97,000$ human ratings for each of the individual dimensions as well as the overall MOS. To formulate the training set for ALLD, we utilize the LLaMA3.1-70B-Instruct model to generate a total of $20k$ training examples for MOS prediction (10k) and A/B test (10k), which includes $2,322$ speakers based on the largest subset NISQA_TRAIN_SIM. Meanwhile, NISQA_TRAIN_SIM with 938 speakers are constructed as a 5k in-domain test set for these two tasks. For MOS prediction, the average response lengths of the training and evaluation set are both. For the A/B test, their average response lengths increase to $42.6$ and $43.3$ respectively. For each example, 5 votes are dedicated to scoring overall MOS, Noisiness, Coloration, Discontinuity, and Loudness. Additionally, the NISQA_VAL_LIVE, NISQA_Test_FOR, and NISQA_TEST_P501 are used for out-of-domain evaluation, containing unseen speech samples from various domains, as summarized in Table 2. For SWD tasks, we utilize LibriSpeech for data generation, with further details provided in Appendix D.

**Models and Baselines**. For MOS prediction, regression models CNN-SA-AP (Mittag et al., 2021), Wav2vec2 (Baevski et al., 2020), and WavLM (Chen et al., 2022) are employed as baseline that only estimate the score without analysis. The former is the SOTA on the NISQA dataset, while the latter two are widely used self-supervised learning models for MOS estimation. For audio LLMs, we utilize the SALMONN (Tang et al., 2023), Qwen-Audio (Chu et al., 2023), and Qwen2-Audio (Chu et al., 2024). Each audio LLM is representative in its own way: SALMONN can extract more acoustic information via bi-encoders, and connect them to LLMs via a Q-former, with LoRA integrated. Qwen-Audio makes the encoder trainable while freezing the entire LLM. In contrast, Qwen2-Audio enables full end-to-end training of both the encoder and the LLM.

**Training Detail.** Besides full parameter finetuning, we also adopt parameter-efficient finetuning for these audio LLMs including IA3 (Liu et al., 2022) (apply to all linear layers) and LoRA (Hu et al., 2021). LoRA matrix adds all queries, keys, and values into the encoder and LLM with a rank of 16. For ALLD, $\beta$ is set as $0.4$ to enhance the distillation, and the learning rate is set as 5e-6. Half of the training examples are used for warm-up finetuning, and then perform sampling on the whole training set to construct a comparison dataset $D$.

**Evaluation Metric**. For MOS numerical prediction, we employ linear correlation coefficient (LCC), Spearman's rank correlation coefficient (SRCC) and mean square error (MSE) as evaluation metrics. Then BLEU score is used to measure the quality of descriptive analysis. Then the For A/B test, in addition to BLEU, we count the accuracy (Acc) to evaluate whether the model provides correct judgement. Since the response is natural language, we further employ a 70B LLaMA-3.1 model to extract the result for Acc calculation. More details of instruction prompt are in Appendix B. Accuracy is also used for SWD evaluation.

### 5.2 RESULT ON MOS PREDICTION

**Main result.** We first report the results of MOS prediction in Table 1, including BLEU, LCC, SRCC, and MSE calculated from regression models (ID 1–3) and audio LLMs with different tuning manners (ID 4–13). It is noted that we train Q-former and LoRA (in LLM part) for "SALMONN", and train the Encoder module with projection layer for "Qwen-Audio". These tuning approaches are consistent with their original papers, since we intend to explore whether speech quality evaluation can be integrated into their multi-task learning process. For Qwen2-Audio, we investigate various tuning manners as all parameters are trainable during its pre-training. From Table 1, we observe that ALLD achieves the best performance across all systems according to evaluation metrics, and the BLEU score demonstrates the efficacy of this distillation strategy. Furthermore, some noteworthy experimental phenomena and insights are reported as follows:

- Smaller regression models excel in numerical MOS prediction compared with many audio LLMs, while they are unable to provide any descriptive analysis or reasoning. In other words, if an audio LLM is adapted solely for MOS estimation, the majority of its parameters would be underutilized, leading to inefficient use of the model capacity on language.

Table 1: MOS prediction results with LCC, SRCC, MSE, and BLEU. For "SALMONN" and "Qwen-Audio", the tuning manners remain consistent with their multi-task pre-training. "*N.A.*" denotes that regression models can not provide descriptive responses.

| ID | Model | Tuning Manner | LCC ↑ | SRCC ↑ | MSE ↓ | BLEU ↑ |
|----|-------|---------------|-------|--------|-------|--------|
| *Regression Model w/o Description* | | | | | | |
| 1 | CNN-SA-AP | | 0.90 | 0.89 | 0.23 | |
| 2 | WavLM | Full-ft | 0.90 | 0.90 | 0.24 | *N.A.* |
| 3 | Wav2vec2 | | **0.93** | 0.92 | 0.27 | |
| *Audio LLMs w/ Description* | | | | | | |
| 4 | SALMONN-7B | Q-former + LoRA | 0.87 | 0.87 | 0.34 | 25.49 |
| 5 | SALMONN-13B | | 0.87 | 0.87 | 0.33 | 25.07 |
| 6 | Qwen-Audio | Enc + Proj. | 0.88 | 0.87 | 0.26 | 23.52 |
| 7 | | IA3 | 0.25 | 0.24 | 1.45 | 16.79 |
| 8 | | LoRA (Enc & Dec) | 0.75 | 0.74 | 0.52 | 18.80 |
| 9 | Qwen2-Audio | Enc-only | 0.89 | 0.89 | 0.24 | 23.41 |
| 10 | | Dec-only | 0.76 | 0.75 | 0.55 | 19.62 |
| 11 | | Full-ft | 0.91 | 0.90 | 0.21 | 23.84 |
| 12 | | **ALLD** | 0.92 | 0.92 | 0.20 | 25.22 |
| 13 | | **ALLD** (2×) | **0.93** | **0.93** | **0.17** | **25.84** |

Table 2: Performance on unseen speech domains. The subscript numbers between brackets represent the performance changes from *in-domain* performance, where the improved metrics are in green.

| Unseen Speech Domains | Model | LCC↑ | SRCC↑ | MSE↓ | BLEU↑ |
|-----------------------|-------|------|-------|------|-------|
| LIVE: *Phone; Skype recording* | Wav2vec2 | $0.86_{(-0.07)}$ | $0.86_{(-0.06)}$ | $0.14_{(-0.13)}$ | - |
| | ALLD | $0.86_{(-0.06)}$ | $0.86_{(-0.06)}$ | $0.14_{(-0.06)}$ | $26.62_{(+1.40)}$ |
| FOR: *forensic speech dataset* | Wav2vec2 | $0.93_{(-0.00)}$ | $0.92_{(-0.00)}$ | $0.13_{(-0.14)}$ | - |
| | ALLD | $0.94_{(+0.02)}$ | $0.93_{(+0.01)}$ | $0.10_{(-0.10)}$ | $25.98_{(+0.76)}$ |
| P501: *Annex C files from P.501* | Wav2vec2 | $0.94_{(+0.01)}$ | $0.94_{(+0.02)}$ | $0.43_{(+0.16)}$ | - |
| | ALLD | $0.92_{(-0.00)}$ | $0.92_{(-0.00)}$ | $0.19_{(-0.01)}$ | $27.23_{(+2.01)}$ |

- Training an audio Encoder is crucial for audio LLMs to obtain the capacity of speech quality evaluation. The poor performance of the system ID 10 confirms that the audio LLMs were not exposed to this task during their pre-training stage. As a result, the representations extracted by the audio encoder do not include quality-related information.

- Parameter-efficient finetuning also fails to achieve satisfactory performance in MOS prediction accuracy. Systems 7 and 8 can mimic the format of response, but they are observed to produce specious and baseless analysis due to limited learning capacity.

In Table 1, the "2×" denotes that we generate the training set twice (total 20k) by modifying the in-context learning demonstrations and adjusting the temperature $\tau$ during LLM inference. This strategy is initially expected to enhance the diversity of descriptive analysis. However, the MSE metric of ALLD surprisingly reduces from $0.20$ to $0.17$. This performance gain is counter-intuitive since the extra 10k of training examples do not introduce more labelled MOS values. We attribute this phenomenon to the impact of the descriptive analysis, which influences the MOS value estimation for audio LLM in a CoT manner. More discussion is in Appendix A.

We then report the performance of system ID 12 on the unseen speech domain in Table 2, including the test sets of LIVE, FOR and P501. The best regression model Wav2vec2 is employed for comparison. It is observed that the model's learned ability to evaluate speech quality generalizes well to unseen speech domains, under the same scoring system. For both MSE and BLEU, the performance even becomes better when encountering domain mismatch.

Table 3: A/B test results *w-* and *w/o* joint training of MOS prediction. It indicates that joint training provides performance gain for A/B test while hardly leading to degradation for MOS prediction.

| Joint Training | A/B Test | | MOS | | | |
|---|---|---|---|---|---|---|
| | BLEU | Acc (%) | LCC | SRCC | MSE | BLEU |
| ✗ | 29.02 | 95.6 | 0.92 | 0.92 | 0.20 | 25.22 |
| ✓ | 30.17 | 98.6 | 0.92 | 0.91 | 0.20 | 26.08 |

Table 4: SWD Accuracy (%) results on LibriSpeech.

| #Syn. words | Length | Random | VC-330 | VC-830 | SSR |
|---|---|---|---|---|---|
| 1 | 14.8 | 6.76 | 51.72 | 49.70 | 45.84 |
| ≥2 | 14.0 | 7.69 | 44.83 | 47.27 | 41.82 |

**Ablation study.** We analyze the effect of ALLD compared with full-ft, which is the best baseline presented in Table 1. To examine the efficacy of distillation, we vary the amount of training examples from 1k to 20k data points, and the histogram of MSE and BLEU results are shown in Fig. 3. We observe that both full-ft and ALLD improve consistently with more data, while with token-level distillation, ALLD shows extra performance gain on BLEU compared to full-ft.

### 5.3 RESULT ON A/B TEST AND SYNTHETIC WORD DETECTION (SWD)

We report the A/B Test results in Table 3. Considering that descriptions in the MOS prediction task benefit the comparison of speech quality, we evaluated the performance of using 10k A/B test data alone (*w/o* MOS) and joint training with both tasks (*w-* MOS). The results demonstrated that joint training can improve A/B test judgment in terms of both BLEU score and accuracy. Meanwhile, the A/B test has a negligible effect on MOS numerical estimation but does contribute to an increase in BLEU score. We have more discussion about joint training in Appendix A.

Considering SWD is not an open-ended task, we report the accuracy with different synthetic words in Table 4. The average length of utterance is added for reference, as well as the accuracy of a random guess. We observed that (i) By comparing the results of the three models, we observe a correlation between detection accuracy and model performance. From VC-330 to VC-830 and finally, to SSR, the detection accuracy gradually decreases, while the subjective MOS reported in their original papers progressively improves. (ii) Multiple modified words have lower detection accuracy, though they are successive in one utterance. This is attributed to the fact that the majority of samples in our training set involved modifications to only a single word. Additionally, human listeners are nearly unable to perform this task, since they cannot distinguish between the original and edited speech by VoiceCraft (Peng et al., 2024).

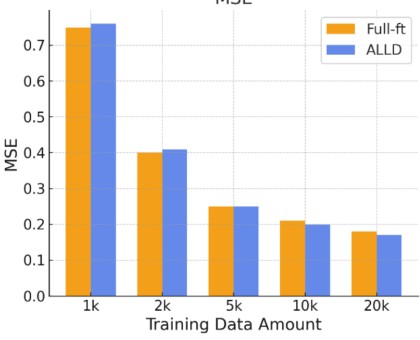 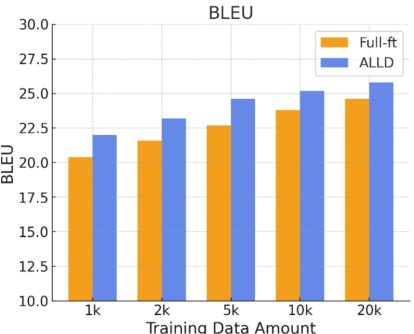

Figure 3: Comparison results between full-ft and ALLD with different data amounts. The LCC and SRCC results show strong linear relations with MSE.

## 6 CONCLUSION

In this work, we aim to teach audio LLMs to perceive and evaluate speech quality with descriptive details. To achieve this, we introduce a speech evaluation corpus that offers multi-dimensional analysis, generated by LLMs using authentic human-annotated scores. We also propose ALLD, a token-level distillation method designed to improve the quality of audio LLM outputs. Experimental results show that ALLD outperforms traditional regression models in terms of LCC, SRCC, and MSE on both MOS prediction and A/B testing tasks, while also generating meaningful responses with a BLEU score of 25.8. Our approach would represent an initial step towards developing a physically-aware intelligent model capable of understanding real-world auditory sensory inputs.

## ACKNOWLEDGMENT

This work has been supported by the National Research Foundation Singapore under AI Singapore Programme (Award Number: AISG2-TC-2022-004). This work also credits the technical supports from NVIDIA Taiwan Research & Development Center (TRDC).

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

## A ADDITIONAL DISCUSSION AND LIMITATION

**The relationship between descriptive analysis and numerical MOS estimation.**

To validate this issue, we retrain a model using the 10k results with descriptive analysis removed. The model's output is set as a fixed template: *"The overall MOS is:"*. All other training hyperparameters remain consistent. Surprisingly, it achieved comparable performance to ALLD, with an MSE of 0.20, and LCC and SRCC of 0.92. When we introduced descriptive diversity to ALLD, the MSE further decreased to 0.17, indicating that the generated responses by the audio LLM do indeed influence the final numerical prediction. We also conducted a case study to verify this: for the same audio sample, when we manually alter the generated description (e.g., changing *"with significant background noise"* to *"without any background noise"*), the final MOS estimation is increased due to the modification. The connection between descriptive analysis and MOS estimation requires further exploration, such as using attention maps, task-activating prompting, or other metrics for analysis and quantification. We consider this as part of our future work.

**Speech quality evaluations and continuous joint training.**

Given the massive resource consumption, recalling all task data for audio LLMs during the pretraining stage is impractical. We aim to confirm that quality evaluation, as an orthogonal task, does not interfere with other tasks. To validate this, we selected several fundamental speech tasks for joint training, as shown in Table 3, including an ASR task on CommonVoice (Ardila et al., 2019), speaker-related age and gender prediction tasks on Fair-Speech (Veliche et al., 2024), and a nonspeech automatic audio captioning task on (Drossos et al., 2020). We found that when training these tasks together on the 7B Qwen-Audio2 model, there was no obvious degradation in MOS prediction performance, with an MSE of 0.25, an LCC of 0.92, SRCC of 0.91.

**Limitation and future directions.**

Both MOS prediction and A/B test tasks are conducted at the utterance level, but in reality, many instances of speech quality degradation occur at the word level. This is particularly true for current zero-TTS models, where only a few words in a sentence might contain defects. However, due to the lack of fine-grained labels, the proposed SWD task can only detect synthetic words but cannot explain the underlying reasons for the degradation. Therefore, audio LLMs are still unable to precisely describe the problematic words or segments in the speech like humans. We believe that this area requires further research to develop a more nuanced mapping between speech signals and language, enabling audio LLMs to perceive and understand speech signals comprehensively and thoroughly.

## B DETAILS OF DATA GENERATION

**MOS prediction.** Given a tuple of metadata of {mos, noi, col, dis, loud}, the generation template for LLaMA-3.1 70B is shown as follows:

*I will give you a tuple of meta information for speech quality evaluation, it contains 5 factors are rating from 1 to 5. For all these factors, higher is better.*

*(1) mos: the overall quality. 1 is very bad, 2 is poor, 3 is fair, 4 is good, 5 is excellent.*
*(2) noi: the level of noise in the audio, reflecting the impact of background noise or other non-speech interference on audio quality. 1 is very noisy, 2 is somewhat noisy, 3 is neither noisy nor clean, 4 is somewhat clean, and 5 is completely clean.*
*(3) col: the alterations in the natural sound of speech caused by distortions or unwanted modifications. 1 is severely distorted, 2 is significantly distorted, 3 is moderately distorted, 4 is slightly distorted, and 5 is no distortion.*
*(4) dis: the discontinuity in the audio, reflecting whether there are breaks, stutters, or incoherence during playback. 1 is severely discontinuous, 2 is significantly discontinuous, 3 is moderately discontinuous, 4 is slightly discontinuous, and 5 is no discontinuity.*
*(5) loud: the perceived volume or loudness of the audio. 1 is extremely quiet, 2 is significantly quiet, 3 is soft but understandable, 4 is clearly loud, and 5 is perfectly loud.*
*I need you to generate a descriptive evaluation for this speech, including a description according to the score from (2) to (5), analyze how they influence the overall quality, and add the mos in the end. For example, input is {demonstration data point}, then you should output: {customized response}*
*· · · · · ·*
*Now the input is {current data point}. Please only output the evaluation*:

This template is not the only option, and the above is provided as a reference. During the first generation, we used the default inference parameters of LLaMA-3.1. For the second generation, we adjusted the temperature to 1.1 and set *top_p* to 0.9 to encourage greater diversity.

**A/B test.** The introduction part of the prompt is consistent with MOS prediction. After introducing the sub-dimensions, the prompt becomes:

*I need you to perform A/B test according to their mos (mos higher means winner. You can flexibly select 1 3 aspects from (2) (5) with an obvious gap (usually score difference more than 0.5), then compare them according to these distinctions. Finally, please give your preference with a reasonable analysis.*

Then, we use LLaMA-3.1-70b to summarize the comparing result from audio LLM generation using the following prompt template:

*"According to the context, please judge if SpeechA is better or SpeechB is better. Only output '[SpeechA]' or '[SpeechB]', do not give any analysis."*

We use this template to extract the answer of better speech (i.e., SpeechA or SpeechB) from both audio LLM generations and the ground-truth transcription, and then judge their consistency to calculate the final accuracy metric. Specifically, we use this two-stage strategy to calculate accuracy because LLaMA-3.1-70b fails to complete it no matter how we design the prompt.

# C  IN-CONTEXT LEARNING ABILITY OF AUDIO LLMS

We evaluate the in-context learning ability of multiple popular open-sourced speech/audio LLMs, in terms of our investigated speech quality evaluation task, including Qwen2-Audio (Chu et al., 2024), SALMONN (Tang et al., 2023), SpeechGPT (Zhang et al., 2023), AnyGPT (Zhan et al., 2024), SALM (Chen et al., 2024). Unfortunately, all of them fail to reliably evaluate the input speech given our detailed prompt, even producing hallucinations like the example below:

*"User: Please evaluate the noisiness of the speech by predicting 'clean' or 'noisy'. For example: [audio1] is noisy, and [audio2] is clean. [audio3] is"*

*"LLM: This speech is spoken by a man instead of a woman because the voice is relatively low."*

In addition, we mention that the latest work UniAudio-1.5 (Yang et al., 2024a) is the first to investigate the in-context learning ability of audio LLMs. However, this audio LLM has not been publicly available, and their investigated tasks are much simpler than quality evaluation.

## D   DATA GENERATION PIPELINE FOR SWD

Due to the lack of datasets or benchmarks in synthetic word detection (SWD) task, we generate the training and evaluation data based on the speech-evaluation pairs from *train-clean-360* subset of LibriSpeech (Panayotov et al., 2015) corpus as follows:

- Step1: We randomly modify 1 to 3 successive words of the transcription by pre-trained LLMs, while the modified transcription should maintain the correct linguistic and grammatical structure. To reduce the potential bias on modification preference, two pre-trained LLMs are employed for this process, namely LLaMA-3.1-70B-Instruct [1] and Qwen2-72B-Instruct [2]. The modified words are recorded as labels for the subsequent training process.

- Step2: We utilize three pre-trained speech editing models, i.e., VoiceCraft-330M, VoiceCraft-830M, and SSR-Speech (Wang et al., 2024b), to edit original speech based on modified transcription. Notably, with force alignment processing, only the modified words are synthetic, while the rest of the speech remains unchanged. Then, we obtain the pairs of modified transcriptions and edited speech.

- Step3: For each training example, the input is: "*Which n words as synthetic in [audio]?*", where *n* is the number of synthetic words, and *[audio]* is the modified speech samples. The expected output of audio LLM should be these words without the transcribing process.

To obtain the results in Table 4, the Qwen-Audio2 is trained with 30k examples, they are different utterances from *train-clean-360* that are uniformly modified by 2 LLMs and 3 speech editing models. 88% of the examples involve modifying a single word, while 8% and 4% involve modifying two and three words, respectively. Then the model is evaluated on a 3k (500×6) test set, where the source speech is from *dev-clean* subset.

---

[1]https://huggingface.co/meta-llama/Llama-3.1-70B-Instruct
[2]https://huggingface.co/Qwen/Qwen2-72B-Instruct

