# OpenReview forum: "Audio Large Language Models Can Be Descriptive Speech Quality Evaluators"
_ICLR.cc/2025/Conference — ICLR 2025 Poster_

### Official Review · Reviewer_LNjn · 2024-10-31

**Soundness:** 3
**Presentation:** 2
**Contribution:** 3
**Rating:** 6
**Confidence:** 2

**Summary:**

This paper introduces a new approach for the speech quality evaluation tasks using audio llm. Additionally, the paper also introduce a new dataset for speech quality evaluation that include human like natural language description of the quality of speech. The proposed method alignment with LLM distillation (ALLD) for the audio LLM training. The ALLD aligns the generated sequence of the audio LLM to an expert LLM's response based on meta-information. The author also evaluated the proposed ALLD model on several evaluation dataset. The experiments showed that the proposed method outperformed SOTA regression model.

**Strengths:**

1. The paper introduces a new dataset for speech quality evaluation using human like natural language description.
2. The proposed ALLD seems like an interesting approach which leverages token level distillation based on the implicit reward modeling from DPO.
3. The paper also shows that the speech quality evaluation can be integrated into multitask learning framework without degrading the performance of other tasks.
4. The experiment results look promising

**Weaknesses:**

1. The author might need give better clarification of the dataset. At the beginning, I had the impression that the corpus was newly created from the paper. Actually the dataset is from previous work but generated the human like description labels from LLM.
2. The proposed method showed better results than Full-ft method. However, the improvement doesn't look significant.

**Questions:**

1. The formulate 1 looks like it is based on token level. However the formulate 2 is sequence level. How is formulate 1 rewritten into formulate 2?
2. why the proposed ALLD gave improved results over Full-FT? Could you share more insights? Is the improvement from the newly prepared data corpus? or the proposed ALLD method?

---

> ### Author Response · Authors · 2024-11-16
> **Response to Reviewer LNjn**
>
> Thank you for the time and effort you've put into reviewing. Based on your valuable feedback, we have summarized the following issues and responded to each one to address your concerns.
>
> Q1: The question about formulate 1 and formulate 2.  \
> Thanks for your question. We would clarify that formulate 2 is also token-level. The probabilities of entire sequences $\pi (y|x)$ are products of token-level probabilities. When we apply the logarithm, the product of probabilities becomes a sum of log probabilities for each token. Expanding formulate 2 using this role, the KL divergence is a token-level constraint applied to each token's probability difference between the model and the reference, accumulated over the sequence.
>
> Q2: Why does the proposed ALLD give improved results over Full-FT?  \
> The benefits brought by ALLD stem from two main reasons:  \
> (1) Token-level distillation. We observed that after SFT, the diversity in response language from the audio LLM was not sufficient, particularly in A/B testing tasks. This was mainly due to insufficient data and the comparatively weaker language description ability of audio LLMs compared to standard LLMs. Therefore, token-level distillation uses an external LLM to further enhance its descriptive capabilities, improving the quality of the generated descriptions.  \
> (2) The exposure bias problem. Due to the teacher-forcing algorithm, the model predicts the current token based on ground-truth sequence during training, while having to depend on its own predicted sequence during inference due to the unavailability of ground truth. This mismatch results in exposure bias, and the RL-based ALLD approach contains a sampling process that can alleviate this issue, and is widely used in sequence generation tasks [1-3]. \
> Additionally, the motivation of this work is that we found mainstream audio LLMs can not perceive all details of the audio and perform speech quality evaluation, due to the lack of suitable training corpus. Based on this, we introduce this corpus based on the existing human rating dataset, which is suitable for audio LLM learning.
>
> We would be delighted to receive any additional suggestions or comments during discussion phase.
>
> Reference
>
> [1] Bahdanau D, Brakel P, Xu K, et al. An actor-critic algorithm for sequence prediction[J]. arXiv preprint arXiv:1607.07086, 2016. \
> [2] Rennie S J, Marcheret E, Mroueh Y, et al. Self-critical sequence training for image captioning[C]//Proceedings of the IEEE conference on computer vision and pattern recognition. 2017: 7008-7024. \
> [3] Prabhavalkar R, Sainath T N, Wu Y, et al. Minimum word error rate training for attention-based sequence-to-sequence models[C]//2018 IEEE International Conference on Acoustics, Speech and Signal Processing (ICASSP). IEEE, 2018: 4839-4843.

---

### Official Review · Reviewer_u8Um · 2024-11-03

**Soundness:** 3
**Presentation:** 3
**Contribution:** 3
**Rating:** 8
**Confidence:** 3

**Summary:**

This paper presents an innovative approach to integrating speech quality evaluation into audio large language models (audio LLMs), addressing a gap in audio LLMs that typically overlook signal quality. The authors introduce a novel dataset that pairs Mean Opinion Score (MOS) ratings with natural language-based assessments (text descriptions) across multiple dimensions of audio quality, including noisiness, coloration, discontinuity, and loudness. This allows the audio LLM to generate descriptive judgments of quality. They propose an "Alignment with LLM Distillation" (ALLD) framework, distilling knowledge from a reference LLM to guide the audio LLM in emulating human-like quality assessments. Experimental results show that ALLD outperforms traditional MOS prediction models in both accuracy and descriptive capability, achieving lower mean square errors and higher BLEU scores.

**Strengths:**

**Originality**: This work introduces a descriptive, language-based dataset for speech quality assessment, allowing audio LLMs to conduct more nuanced evaluations. The ALLD framework presents an innovative approach by guiding audio LLMs through token-level distillation.


**Quality**: The comparisons between traditional regression methods and descriptive, audio LLM-based approaches offer valuable insights into fine-tuning and demonstrate the effectiveness of natural language guidance.


**Clarity**: The paper is generally well-written and clear, though Section 2.2 could benefit from further refinement.


**Significance**: Repurposing Audio LLMs for speech quality evaluation addresses a critical need, given the limited availability of reliable automated metrics and the high costs of subjective evaluation. This work provides a pr

**Weaknesses:**

* While a strengths and weaknesses comparison between two systems across specific sub-dimensions is reasonable, it is unclear how a human or LLM might synthesize these into an overall preference judgment. For example, if System 1 outperforms System 2 in one sub-dimension but falls behind in another, the basis for an overarching preference remains ambiguous. The dataset relies on LLM responses to make arbitrary decisions on whether Speech A or Speech B is preferable, using sub-dimensional scores as part of the reasoning. This raises concerns about consistency and the interpretability of such comparative judgments.
* Since the Teacher model is also used as the description generator, it would be useful to explore the generalizability of the ALLD framework. Given that BLEU scores are calculated on synthetic descriptions produced by the Teacher, examining how well ALLD performs on independently generated descriptions could add valuable insights. For example, one could compare ALLD with a smaller set of human-generated descriptions on the same dataset to assess its generalizability further.

**Questions:**

1. Can the authors kindly clarify the framework or the decision criteria for generating preference judgments when systems perform differently across sub-dimensions?
2. If the teacher model used during distillation is the same as the description generator of the training dataset, could this lead to a potential bias in the interpretation of the BLEU scores?
3. Is there empirical support for the claim that human listeners cannot distinguish between original and edited speech produced by VoiceCraft? Including a perceptual test comparing original and VoiceCraft-edited audio samples or citing relevant literature if such a test exists can help here.

---

> ### Author Response · Authors · 2024-11-16
> **Response to Reviewer u8Um**
>
> Thank you for the time and effort you've put into reviewing. Based on your valuable feedback, we have summarized the following issues and responded to each one to address your concerns.
>
> Q1:  The decision criteria for generating preference judgments.  \
> To faithfully clarify,  the proposed our method does NOT require preference judgment. In ALLD, the description generated by the LLM based on meta information is always considered a positive example, as the meta information is derived from the ground truth of NISQA, and the LLMs’ descriptive ability is accurate and strong. Conversely, the description produced by the audio LLM from raw audio is treated as a negative example, as it must extract information from the raw audio, making it impossible to be more precise than the ground truth-based description. Thus, although our method is inspired by DPO, it can skip the preference judgment step. A similar setting can be found in SpeechAlign [1].
> In addition, We would like to clarify that ALLD is not a form of "preference optimization." Instead, we found that the audio LLM exhibits some shortcomings (e.g., lack of descriptive diversity) when learning quality evaluation, particularly in A/B test tasks. Therefore, our goal with ALLD is to align the output of the audio LLM with that of the LLM, rather than selecting a better one between them. By leveraging token-level distillation, ALLD were able to achieve this alignment effectively.
>
> Q2: A potential bias in the interpretation of the BLEU scores. \
> Thank you for your insightful question. The model we used for the generation was Llama 3.1-70B, as we found it produced the highest quality among models of 70B size. For distillation, we used Qwen2-7B—primarily to save memory during training, and also because the tokenizers of Qwen2 and Qwen-Audio2 are inherently aligned, which facilitates token-level distillation.
> Additionally, we also experimented with using GPT-4 to generate the descriptions and found that the generated descriptions were similar to those of Llama 3.1, likely because we included the scoring criteria and in-context learning examples in the input.
>
> Q3:  Empirical support for "Human listeners cannot distinguish between original and edited speech produced by VoiceCraft."  \
> Thank you for your question. We refer to the results in paper [2]. In Figure 1, human listeners preferred VOICECRAFT-edited speech over the original real recording 48% of the time in side-by-side naturalness comparisons. Specifically, in the 1-2 word results shown in Figure 3, VOICECRAFT accounted for **39.68%**, raw speech for **40.95%**, and Neutral for 19.37%. Since most of the edits in our work only contain a single word, this result provides meaningful evidence for this view.
>
> We would be delighted to receive any additional suggestions or comments during discussion phase.
>
>
> Reference  \
> [1] Zhang D, Li Z, Li S, et al. SpeechAlign: Aligning Speech Generation to Human Preferences[J]. arXiv preprint arXiv:2404.05600, 2024.  \
> [2] Peng, Puyuan, et al. "Voicecraft: Zero-shot speech editing and text-to-speech in the wild." arXiv preprint arXiv:2403.16973 (2024).

---

> ### Comment · Reviewer_u8Um · 2024-11-22
> **Official Comment by Reviewer u8Um**
>
> Thank you for your detailed and thoughtful responses to my queries.
> I have updated my score based on these responses and commend the authors for their work. I look forward to seeing these additional details incorporated into the paper for further clarity and transparency.

---

> > ### Author Response · Authors · 2024-11-22
> > **Reponse to Reviewer u8Um**
> >
> > Thank you for improving the rating, and we would like to express our gratitude once again for your valuable feedback.

---

### Official Review · Reviewer_eNi6 · 2024-11-03

**Soundness:** 3
**Presentation:** 4
**Contribution:** 4
**Rating:** 8
**Confidence:** 4

**Summary:**

The paper contributes to the below 3 aspects and talks importance of descriptive information of audios in Multimodal LLM
(i) Importance of speech quality evaluation in multimodal agents especially for speech. (ii) Provide descriptive speech quality evaluation sets important for benchmarking the evaluation of audio quality in Audio LLMs. (iii) A novel learning strategy called ALLD that allows end to end perception and generation

**Strengths:**

The paper shows clear outcomes for the novel proposal and experiment
- ALLD achieves the best performance across all systems according to evaluation metrics, and the BLEU score demonstrates the efficacy of this distillation strategy
- Paper describes means of generating evaluation data which is descriptive and can improve Audio LLM performance

**Weaknesses:**

While the improvements from the experiments have shown improvement its not clear on why the LCC and SRCC haven't improved for LIVE and P501 datasets.
Also, descriptive language is subjective to users, unlike evaluation score like BLEU, how do you propose to adhere to similar descriptive style for the evaluation generation

**Questions:**

Please explain the reason for lower LCC and SRCC for LIVE and P501 datasets.

Please share details on how you plan to generate descriptive text for audio data maintaining uniformity

---

> ### Author Response · Authors · 2024-11-16
> **Response to Reviewer eNi6**
>
> Thank you for the time and effort you've put into reviewing. Based on your valuable feedback, we have summarized the following issues and responded to each one to address your concerns.
>
> Q1: Explain the reason for lower LCC and SRCC for LIVE and P501 datasets.   \
> Thank you for your question. The lower LCC and SRCC are compared with their in-domain (test-sim) performance, primarily because the speech from these two subsets was not included in the training set. Since their speech domains are different, a slight drop in performance is inevitable.
>
>
> Q2: How to generate descriptive text for audio data maintaining uniformity. \
> Thank you for raising this important topic. We would like to share some observations and insights to address your concern: \
> (1) The demonstrations for in-context learning are useful for uniformity. As mentioned on line 205, we provide several golden examples (e.g., example in line 072) to guide the LLM in understanding the expected format of the generated responses. Additionally, the text-based scoring criteria are provided as input to the LLM for reference. As a result, the language style generated by the same LLM tends to be consistent. \
> (2) We have also considered the issue of subjectivity in language descriptions. Initially, our approach was to use different LLMs for the descriptions to simulate linguistic diversity. However, we found that even though this task may seem straightforward, only Llama 3.1 (among models of around 70B parameters) performed well enough. GPT-4 also produced high-quality text, but due to cost considerations, we ultimately opted for an open-source model. \
> (3) In our subsequent research, we design a mechanism to increase sample encourage diversity in the generated responses by adjusting the top-p and temperature factor during LLM inference. We generated a total of five variations to simulate five different annotators. The results of MSE, LCC and SRCC metrics remained largely consistent with previous results (0.17, 0.93, 0.93), while BLEU scores showed a slight improvement (average 26.4 on 5 test sets).
>
> We would be delighted to receive any additional suggestions or comments during discussion phase.

---

### Official Review · Reviewer_XRes · 2024-11-06

**Soundness:** 3
**Presentation:** 3
**Contribution:** 2
**Rating:** 5
**Confidence:** 4

**Summary:**

This paper focuses on predicting MOS using a recently popular audio LLM approach. The model provides a natural language description of audio quality across several key dimensions, such as noisiness, coloration, and discontinuity, before assigning an overall MOS score. This process resembles a CoT approach, in which the model evaluates different aspects of audio quality to arrive at a final score. The authors claim that this method of MOS prediction is more accurate than traditional regression-based MOS prediction models.

**Strengths:**

This paper proposes to predict the MOS within the framework of a currently popular audio LLM scheme. The authors present their method clearly and effectively, detailing how the model assesses various aspects of audio quality before generating an overall MOS score. Experimental results demonstrate the effectiveness of this approach, indicating that the proposed method achieves a higher accuracy in MOS prediction compared to traditional regression-based methods.

**Weaknesses:**

- According to the ITU-T definition, the MOS should be an integer between 1 and 5. However, in the example provided in Section 1, the sentence ".....Taking into account all factors, the overall MOS score is only 2.4" conflicts with this definition, as MOS should not be a decimal. This discrepancy suggests a fundamental misalignment with the official MOS standard, which could impact the validity of the work.
- Although the authors extend the application of audio LLMs to MOS prediction, this appears to be a relatively incremental extension of prior works, such as SALMONN, which already covers a wide range of audio understanding tasks. As such, the methodological contribution may seem limited, potentially lacking the level of innovation typically expected at ICLR.
- The authors mention in Section 4.3 that the model requires SFT before RL. However, SFT is not mentioned at all until this point, which may lead readers to assume that RL alone is responsible for training the model. A more transparent approach would be to explicitly state from the outset that the model uses a typical SFT + RL pipeline, rather than emphasizing only certain parts of the process.
- In later sections, the term "full-ft" appears, but it is unclear if this is meant to refer to the same process as SFT. If they are indeed synonymous, it would be clearer and less confusing to use consistent terminology throughout the paper. This will ensure readers understand the training pipeline without ambiguity.

**Questions:**

Please refer to the weakness part.

---

> ### Author Response · Authors · 2024-11-16
> **Resonse to Reviewer XRes**
>
> Thank you for the time and effort you've put into reviewing. Based on your valuable feedback, we have summarized the following issues and responded to each one to address your concerns.
>
> Q1: the MOS should be an integer between 1 and 5. \
> Thank you for your question. In the MOS prediction task (dataset), the ground truth MOS is the **average score given by multiple listeners**. This setting stems from the subjective variation of human perception, the MOS scores for the same sample could vary, thus averaging these scores provides a more objective reflection of speech quality. Moreover, **almost all existing MOS prediction works** [1] focus on predicting the decimal MOS score. Therefore, aligning with these works provides quantitative references and reproducible comparison.
>
> Q2:  Lack of Novelty: Prior works, such as SALMONN, have already covered a wide range of audio understanding tasks. \
> As we highlighted in the Abstract (line 14), speech quality evaluation has been ignored in these audio LLMs, which would be due to the lack of suitable datasets. Additionally, in the main text (line 225) and Appendix C, we demonstrated that even with in-context learning, a series of audio LLMs are **not** able to evaluate speech quality, including SALMONN. Based on this, we introduce a dataset suitable for audio LLM learning and also propose new methods to improve the learning effectiveness. Therefore, both the motivation and contributions of our work are distinct from those of works like SALMONN, and we believe it fits well with ICLR, a conference that encourages innovation.
>
> Q3: ALLD contains STF and RL. \
> From a rigorous perspective, ALLD cannot be defined as a typical RLHF method—we defined it as a “distillation method” to align the behavior of the audio LLM with that of an LLM (line235). We also explain its differences from traditional RLHF (starting from line 243). Apart from the reference model mentioned, the positive samples used in the optimization process are ground-truth rather than sampled by the model itself. In other words, ALLD is essentially an enhanced version of SFT, motivated by the goal of improving the quality of the responses generated by the audio LLM rather than to optimize its generation preferences. \
> Additionally, even in other RLHF methods, RL is not the only needed in RLHF. The importance of using the SFT as an auxiliary module has been explored in [2], and the open-source implementation of DPO includes a hyperparameter “rpo_alpha ”for SFT loss (as seen here: https://huggingface.co/docs/trl/dpo_trainer). Thanks for your question, and we will clarify further in the next version.
>
> Q4: Unclear about "Full-ft" and "SFT". \
> We would like to clarify that “full-FT” and “SFT” are not the same concept. “Full-FT” refers to **fully fine-tuning all the parameters** of the model. It highlights “how many parameters are trainable”,  as listed alongside LoRA tuning and encoder-only tuning in Table 1. On the other hand, “SFT” refers to **supervised fine-tuning with labelled data**, which is a primary method for enabling LLMs to follow instructions. We will add further clarification to avoid confusion.
>
>
> We hope the above discussion has clarified the perspectives the reviewer was interested in exploring further. We would be delighted to receive any additional suggestions or comments.
>
>
>
> Reference \
> [1] Lo C C, Fu S W, Huang W C, et al. Mosnet: Deep learning based objective assessment for voice conversion[J]. arXiv preprint arXiv:1904.08352, 2019. \
> [2] Pang R Y, Yuan W, Cho K, et al. Iterative reasoning preference optimization[J]. arXiv preprint arXiv:2404.19733, 2024.

---

> ### Comment · Reviewer_XRes · 2024-11-22
>
> Thank you for addressing some of my concerns. I understand that previous audio LLM studies did not include audio quality evaluation tasks. However, the methodology for this task seems similar to other tasks, offering limited new ideas and insights. With that being said, I still appreciate the effort of this work. I would like to improve the score to 5, leaving room for improvement to meet my higher expectations for novelty in this work.

---

> > ### Author Response · Authors · 2024-11-22
> > **Response to Reviewer XRes**
> >
> > Thank you for joining the discussion and improving the rating.
> >
> > First, we appreciate your understanding that quality evaluation tasks are a novel topic for audio LLMs. We contribute to this area by introducing a relevant dataset and a learning approach to enable the descriptive evaluation capacity of open-source audio LLMs. The proposed methd ALLD does NOT require any preference annotations, and its "reference model" also plays a completely different role with RLHF. We believe this method is novel, generalizable, and provides valuable insights for other tasks that **require audio LLMs to provide high-quality descriptions from raw audio**. Based on this, we sincerely hope you could reconsider the novelty of this work (since rating 5 denotes boardline reject).  Meanwhile, we look forward to hearing your thoughts on the higher expectations for novelty, as well as your valuable feedback on areas for improvement.

---

### Meta-Review · Area_Chair_twoS · 2024-12-21

**Metareview:**

This paper performs audio quality evaluation (e.g. "naturalness" of speech) with audio large language models (audio LLMs), addressing a gap in audio LLMs that typically overlook signal quality. Authors collected a dataset that pairs Mean Opinion Score (MOS) ratings with natural language-based assessments (text descriptions) across multiple dimensions of audio quality, including noisiness, coloration, discontinuity, and loudness. This allows the audio LLM to generate descriptive judgments of quality. They propose an "Alignment with LLM Distillation" (ALLD) framework, distilling knowledge from a reference LLM at the token level to guide the audio LLM in emulating human-like quality assessments. Experimental results show that existing audio LLMs (e.g. SALMONN) do not do well on the proposed task and that ALLD outperforms traditional MOS prediction models in both accuracy and descriptive capability, achieving lower mean square errors and higher BLEU scores.

Strengths
- the paper performs speech & audio quality estimation with LLMs for the first time (AFAIK) and shows that LLMs also outperform on this task which is typically not included in pre-training
- ALLD is an innovative approach that could generalize to other tasks, natural language guidance is explored effectively
- results are strong, outperforming baselines in regression and prediction tasks
- the paper is well written and makes a compelling argument

Weaknesses
- The generalizability of ALLD could be confirmed further (2nd weakness by reviewer u8Um)
- I don't see a commitment by authors to release data or code, which would increase the potential impact of this work (although the description of label generation seems straightforward)

Overall, reviewers recommend acceptance (based on the novel task, good performance and innovative contributions) and the AC concurs. I feel that authors responded well to reviewer XRes concerns. Main reason to not accept this paper would be to require validation of ALLD on further tasks/ data, which I think is outweighed by the significant application already presented here.

**Additional Comments On Reviewer Discussion:**

Authors mostly clarified some points in the paper which I think would be easy to also include in the manuscript.

---

### Decision · Program_Chairs · 2025-01-22

Accept (Poster)